# Syndrome Metabolic Markers, Fitness and Body Fat Is Associated with Sleep Quality in Women with Severe/Morbid Obesity

**DOI:** 10.3390/ijerph18179294

**Published:** 2021-09-03

**Authors:** Claudia Andrea Vargas, Iris Paola Guzmán-Guzmán, Felipe Caamaño-Navarrete, Daniel Jerez-Mayorga, Luis Javier Chirosa-Ríos, Pedro Delgado-Floody

**Affiliations:** 1Department of Physical Education, Sport and Recreation, Universidad de La Frontera, Temuco 4780000, Chile; claudia.vargas@ufrontera.cl; 2Faculty of Chemical-Biological Sciences, Universidad Autónoma de Guerrero, Chilpancingo de los Bravo 39087, Mexico; pao_nkiller@yahoo.com.mx; 3Faculty of Education, Universidad Católica de Temuco, Temuco 4780000, Chile; marfel77@gmail.com; 4Faculty of Rehabilitation Sciences, Universidad Andres Bello, Santiago 7591538, Chile; daniel.jerez@unab.cl; 5Department Physical Education and Sports, Faculty of Sport Sciences, University of Granada, 18011 Granada, Spain; lchirosa@ugr.es

**Keywords:** morbid obesity, exercise, sleep quality, quality of life

## Abstract

Background: Sleep is an important modulator of neuroendocrine function and glucose metabolism. Poor sleep quality is related to metabolic and endocrine alterations, including decreased glucose tolerance, decreased insulin sensitivity, and increased hunger and appetite. Objective: The aim of the present study was to determine the association between sleep quality with metabolic syndrome (MetS) markers, fitness and body fat of women with severe/morbid obesity. Methods: This cross-sectional study included 26 women with severe/morbid obesity. Fasting plasma glucose (FPG), high-density lipids (HDL-c), triglycerides (TGs), and the metabolic outcomes total cholesterol (Tc) and low-density lipids (LDL-c), systolic (SBP) and diastolic blood pressure (DBP), body composition and fitness were measured. Results: Poor sleep quality showed a positive association with body fat (%) ≥ 48.2 (OR; 8.39, 95% CI; 1.13–62.14, *p* = 0.037), morbid obesity (OR; 8.44, 95% CI; 1.15–66.0, *p* = 0.036), glucose ≥ 100 mg/dL (OR; 8.44, 95% CI; 1.15–66.0, *p* = 0.036) and relative handgrip strength ≤ 0.66 (OR; 12.2, 95% CI; 1.79–83.09, *p* = 0.011). Conclusion: sleep quality is associated with health markers in women with severe/morbid obesity.

## 1. Introduction

Sleep is an important modulator of neuroendocrine function and glucose metabolism, and poor sleep quality is related to metabolic and endocrine alterations, including decreased glucose tolerance, decreased insulin sensitivity, and increased hunger and appetite [1]. Poor sleep quality is related to several other metabolic syndrome (MetS) risk factors, such as higher waist circumference (WC), hypertension, elevated serum triglycerides (TGs), low serum high-density lipoprotein cholesterol (HDL-C), and hyperglycaemia [2]. Furthermore, today, a short sleeping time is very common and sleep duration has been associated with a great number of adverse health effects, including all-cause mortality, weight gain, and incident cardiovascular disease [3]. Therefore, reduced sleep is a risk factor for obesity and cardiovascular disease, and consequently, may contribute to worsening of common obesity complications such as metabolic and cardiovascular diseases and reduce cardiorespiratory fitness (CRF) [4].

In addition, evidence showed a strong relationship between sleep quality and hormonal status, with sleep restriction seeming to impair the hormonal system that regulates energy balance. This involves several hormones, including cortisol, insulin, ghrelin, leptin and melatonin, increasing the risk of diabetes [5] and mortality [6].

Moreover, morbid obesity is associated with an increase in sleep disorders such as obstructive sleep apnoea, affecting sleep quality [7]. In this sense, poor sleep quality is strongly associated with mood disturbance and poor health related to quality of life (HRQoL) among patients with morbid obesity [8]. Furthermore, women with morbid obesity have poorer sleep quality [9]. In this sense, poor sleep quality is associated with an adverse metabolic profile and low fitness [10,11], indicating that sleep may play an important role in health disparities and may represent a risk factor for MetS [12,13]. 

The integration of the concepts of sleep quality and duration and the examination of their combined and independent impacts on health outcomes is a sensible and necessary step in understanding the contribution of sleep as a whole to public health [14]. Accordingly, patients with severe/morbid obesity (i.e., ≥obesity class II) present more adverse MetS factors and higher levels of inflammation [5,6,8], especially women suffering from more severe obstructive sleep apnoea syndrome and worse sleep quality [8]. However, how MetS risk factors are related to sleep quality in women with morbid obesity must be studied deeply. Therefore, the present study aimed to determine the association between sleep quality with MetS markers, such as WC, hypertension, elevated TG, low HDL-C, hyperglycaemia, fitness and body fat in women with morbid obesity.

## 2. Materials and Methods

### 2.1. Study Design

This cross-sectional study included 26 female volunteers selected by convenience, who provided signed written consent for participation. Ten participants had obesity (38.5%) and 16 had morbid obesity (61.5%). The study was carried out in accordance with the Declaration of Helsinki (2013) and was approved by the Ethical Committee of the Universidad de La Frontera, Temuco, Chile (Act 080-21).

The inclusion criteria were (i) 18–60 years of age, (ii) women, (iii) medical authorization for physical testing, and (iv) body mass index (BMI) ≥ 35 kg/m^2^ (i.e., ≥obesity class II). The exclusion criteria were (i) physical limitations preventing the performance of the physical test (e.g., restrictive injuries of the musculoskeletal system), (ii) exercise-related dyspnoea or respiratory alterations, and (iii) chronic heart disease with any worsening in the last month.

### 2.2. Measurements

#### 2.2.1. Sleep Quality Measurements

Sleep quality was assessed using the Pittsburgh Sleep Quality Index (PSQI) [15]. The PSQI is a self-report questionnaire that includes seven component scores: subjective sleep quality, sleep latency, sleep duration, habitual sleep efficiency, sleep disturbances, use of sleep medication, and daytime dysfunction. In the PSQI, subjects rate perceived sleep quality as very good, fairly good, fairly bad, or very bad. These subjective scales are weighted to obtain a global PSQI score that differentiates between good and poor SQ. Their sum builds the global PSQI report, which provides an inverse score, where a score <5 denotes good sleep quality (GSQ) and a score >5 denotes poor sleep quality (PSQ). This scale has been used in previous studies [16] and publications that examined bariatric patients [17]. Conditions associated with PSQ include the use of sleep medications, difficulties in daily living and enthusiasm, and low sleep efficiency.

#### 2.2.2. MetS Markers

The MetS markers were screened using standard criteria [2,18]. After overnight fasting for 10 ± 2 h, all patients underwent a baseline assessment (pre-test) between 08:00 and 9:00 in the morning. All participants arrived at the lab for the extraction of a 5 mL blood sample, in order to determine the MetS outcomes: fasting plasma glucose (FPG), HDL-c, and TG. The systolic (SBP) and diastolic blood pressures (DBP) were measured according to the standard criteria [19]. Blood pressure was measured in the sitting position after 5 min rest. Two recordings were taken, and the mean of the measurements was used for statistical analysis using an OMRON^TM^ digital electronic BP monitor (model HEM 7114, Chicago, IL, United States). Caffeine, exercise, and smoking were avoided for at least 30 min prior to the measurement [20]. WC was assessed using a tape measure graduated in centimetres (Adult SECA^TM^) at the upper hipbone and the top of the right iliac crest, with a non-elastic measuring tape in a horizontal plane around the abdomen at the level of the iliac crest. The tape was snug, but not skin compressing, and it was parallel to the floor. The measurement was performed at the end of a normal expiration [21].

#### 2.2.3. Body Composition and Anthropometric Parameters

Body mass (kg) and body fat (%) were measured using a digital bioimpedance scale (TANITA^TM^, model 331, Tokyo, Japan). Height (m) was measured with a SECA^TM^ stadiometer (model 214, Hamburg, Germany), with subjects wearing light clothing and without shoes. The BMI was calculated as the body mass divided by height squared (kg/m^2^), and then used to estimate the degree of obesity using the standard criteria for obesity class [22].

#### 2.2.4. Six-Minutes Walking Test

The day after the metabolic measurements, the physical condition of participants in both groups was measured by endurance and muscle strength testing. First, a six-minute walking test (6Mwt) was used to estimate CRF. The test was performed in a closed space on a flat surface (30 m long), with two reflective cones placed at the ends to indicate the distance. During the test, participants were assisted with instructions from an exercise physiologist [23].

#### 2.2.5. Handgrip Strength

Which has been used in previous studies [24]. Two attempts were performed, measuring each hand, and the best result from each was selected. As previously, the mean value obtained was taken as the total score [24]. Using these data, we calculated other outcomes such as the HGS relative to BMI (HGS/BMI).

### 2.3. Data Analysis

The statistical analysis of the data was carried out using the statistics programs STATA v.15.0 (StataCorp, College Station, TX, USA). The absolute frequencies were determined for the qualitative variables. The comparison between groups was evaluated using Student’s *t* test. In order to determine the linear relation between the sleep quality score and anthropometric, metabolic and fitness parameters, Pearson’s correlation coefficients were calculated, as well as multiple logistic regression models, determining the odds ratios. Values of *p* < 0.05 were considered statistically significant.

## 3. Results

Table 1 shows anthropometric, clinical-metabolic and fitness parameters according to sleep quality. Fourteen women presented good sleep quality (GSQ; 38.42 ± 12.64 years) and twelve poor sleep quality (PSQ; 41.66 ± 11.38 years). The poor sleep quality group reported a higher BMI (GSQ; 39.62 ± 5.72 vs. PSQ; 45.99 ± 7.61 kg/m^2^, *p* = 0.023) and body fat percentage (GSQ; 46.45 ± 4.28 vs. PSQ; 50.21 ± 4.74%, *p* = 0.044) than the good sleep quality group. In addition, they reported lower CRF (GSQ; 560 ± 71.14 vs. PSQ; 458.33 ± 105.12 m, *p* = 0.007).

Figure 1 shows the relationship between a poor sleep quality score and anthropometric, metabolic and fitness parameters. BMI (rho = 0.40, *p* = 0.03, Panel (A)) and triglyceride levels (rho= 0.44, *p* = 0.02, Panel (B)) were linked positively with poor sleep quality, whereas CRF showed an inverse relationship with poor sleep quality (rho = −0.66, *p* < 0.001, Panel (C)). There was not a significant relationship with another sociodemographic and physical variables (Table 2).

In total, 61.5% of patients presented with morbid obesity, and 34.62% presented ≥4 MetS parameters. There were no significant differences between groups (Table 3). 

Table 4 shows the association between poor sleep quality with MetS markers and physical status. Poor sleep quality reported a positive association with body fat (%) ≥ 48.2 (OR; 8.39, 95%CI; 1.13–62.14, *p* = 0.037), morbid obesity condition (OR; 8.44, 95%CI; 1.15–66.0, *p* = 0.036), glucose ≥ 100 mg/dL (OR; 8.44, 95%CI; 1.15–66.0, *p* = 0.036) and relative handgrip strength ≤ 0.66 (OR; 12.2, 95%CI; 1.79–83.09, *p* = 0.011). 

## 4. Discussion

The aim of the present study was to determine the association between sleep quality with MetS markers (i.e., WC, hypertension, elevated TG, low HDL-C, and hyperglycaemia), fitness and body fat in women with severe/morbid obesity. In the present study, poor sleep quality showed an association with body fat and morbid obesity. Previous evidence showed that lack of sleep is a risk factor for obesity, insulin resistance, and type 2 diabetes [23]; therefore, a short sleep duration and other dimensions of poor sleep quality are associated with body fat [16]. In this context, it was demonstrated in adult subjects that poorer sleep efficiency is related to higher fat mass [24]. In addition, a recent study reported that BMI negatively predicted sleep duration and sleep efficiency in Chinese young adults [25]. Sweatt et al. [26] showed that lower sleep quality is associated with elevated visceral adipose tissue, and poor sleep quality was positively associated with fat mass percentage in Spanish subjects [27]. Similarly, another study reported that fewer hours of sleep may be linked to fat mass index and obesity increase in Korean adults [28]. Another study showed that body composition of subjects with obesity is related to sleep habit changes (i.e., sleep quality and quantity), and the results of this study also indicated that sleep disorders such as obstructive sleep apnoea are associated with sarcopenic obesity, and nocturnal hypoxia is linked to obesity [29]. 

Another important result is that poor sleep quality showed an association with glucose alteration (i.e., glucose ≥ 100 mg/dL) and reported a positive relationship with triglyceride levels. An study showed that the anti-inflammatory cytokine (IL-10) serum levels were significantly reduced in subjects with morbid obesity and obstructive sleep apnoea, and reported a strong correlation with a systemic state of hyperinsulinemia and insulin resistance [30]. Likewise, another study reported that fasting glucose is directly linked to sleep duration; nevertheless, sleep quality was not associated with fasting glucose or 2-h glucose in adults who are overweight/obese [31]. Knutson et al. [32] indicated that sleep quality and duration were predictors of haemoglobin A1c (HbA1c) level, an important marker of glucose alteration/control, in American volunteers with diabetes. Moreover, evidence showed that sleep disturbances may be linked to impaired glucose metabolism [33]. In this context, another study conducted in subjects with type 2 diabetes mellitus showed that poor sleep quality was related to worse control of glycaemia [34]. Another study conducted in women who were overweight or obese reported that the participants who had worse sleep quality obtained significantly higher HOMA2-IR; therefore, the authors concluded that more studies were needed to establish whether enhancing sleep quality improves insulin resistance [35]. The evidence also showed that women have lower sleep quality than men [8]. Chirwa et al. [36] reported that worse sleep quality was linked to higher HbA1c and attendant health complications in pregnant women. In addition, Morselli et al. [37] indicated that sleep loss affects insulin sensitivity, leading to increased diabetes risk. Although there is no complete mechanistic explanation for these observations, they point to a mismatch between circadian rhythm and energy homeostasis that may trigger or increase symptoms of obesity and diabetes [38]. 

Fitness levels were linked to poor sleep quality, and sleep quality is positively associated with greater physical fitness and especially with high levels of upper body strength [39]. Hence, excess fat reduces the total compliance of the respiratory system, increases pulmonary resistance, and reduces respiratory muscle strength [40], which can affect the CRF. In this context, other findings indicated that sleep patterns (i.e., quality and duration) were factors that had a significant influence on physical fitness results in Taiwanese adults. Likewise, a cross-sectional study showed that sleep quality was strongly linked to physical fitness (i.e., grip strength, one-leg standing test, back scratch test and vital capacity) in Chinese adults [41]. Moreover, it was reported that worse sleep quality is related to lower levels of physical fitness and less physical activity in young adults [42]. In this context, another study showed that poor sleep quality was associated with worse CRF in adolescent women [43]. On the other hand, evidence showed that a good CRF is fundamental to decrease sleep problems [44]. Likewise, Lee and Lin [45] reported that young adults with worse sleep quality had lower levels of CRF and muscular endurance. Another study showed that sleep problems were associated with poorer physical fitness in women [4]. Therefore, sleep quality is an important marker related to health in different dimensions.

### Limitations

The main limitation of this study was that the participants’ lifestyle such as physical activity levels and eating habits were not measured. Moreover, menopausal status and transition were not considered, which can have an impact on sleep quality and duration.

## 5. Conclusions

In the present study, poor sleep quality in women with severe/morbid obesity showed a negative association with body fat, metabolic outcomes and fitness. For this reason, poor sleep quality represents an important factor that can further aggravate the health of women with morbid obesity. It is necessary to integrate the concepts of sleep quality and duration to examine their combined and independent impacts on health outcomes in order to understand the contribution of sleep as a whole to public health.

## Figures and Tables

**Figure 1 ijerph-18-09294-f001:**
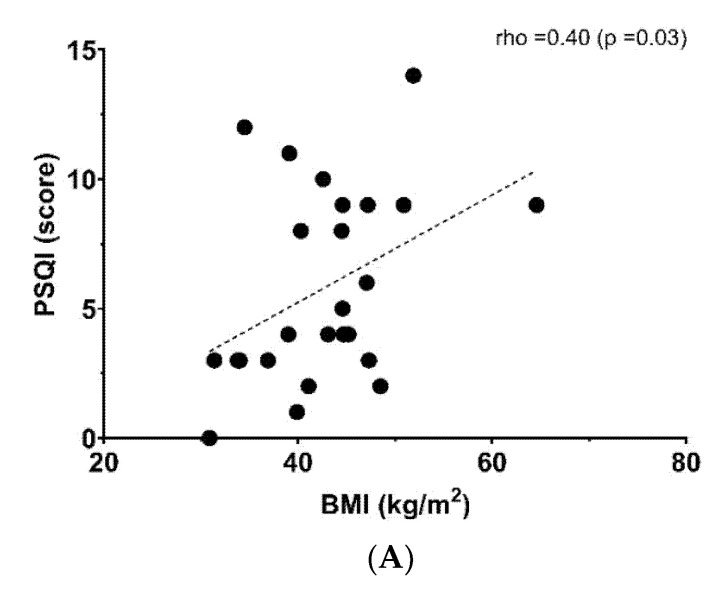
Relationship between poor sleep quality score and (**A**) body mass index (BMI), (**B**) triglycerides (TGs) and (**C**) six-minute walking test (6MWt).

**Table 1 ijerph-18-09294-t001:** Anthropometric, clinical-metabolic and fitness characteristics of the study population according to sleep quality.

		Sleep Quality		
Parameters	Total (*n* = 26)	Good (GSQ, *n* = 14)	Poor (PSQ, *n* = 12)	*p*-Value
Age (y)	39.9 ± 11.94	38.42 ± 12.64	41.66 ± 11.38	0.50
Body mass (kg)	105.1 ± 19.85	98.95 ± 17.93	112.24 ± 20.3	0.60
Height (m)	1.56 ± 0.06	1.57 ± 0.06	1.56 ± 0.06	0.08
BMI (kg/m^2^)	42.56 ± 7.3	39.62 ± 5.72	45.99 ± 7.61	0.023
Waist circumference (cm)	117.2 ± 17.8	113.24 ± 12.35	121.73 ± 14.41	0.11
Body fat (%)	48.2 ± 4.8	46.45 ± 4.28	50.21 ± 4.74	0.044
Clinical-Metabolic				
SBP, mmHg	137.57 ± 16.1	136.85 ± 17.61	138.41 ± 14.86	0.81
DBP, mmHg	87.65 ± 12.03	89.85 ± 11.21	85.08 ± 12.92	0.31
Glucose, mg/dL	99.53 ± 18.96	97.35 ± 21.79	102.08 ± 15.58	0.53
TG, mg/dL	121.15 ± 62.2	103.21 ± 38.0	142.08 ± 78.88	0.11
HDL-c, mg/dL	51.15 ± 10.6	51.92 ± 12.69	51.91 ± 8.07	0.99
Fitness				
6MWt (m)	513.07 ± 100.82	560 ± 71.14	458.33 ± 105.12	0.007
Handgrip strength (kg)	27.86 ± 7.63	28.14 ± 7.55	27.54 ± 8.04	0.84
Relative hand grip strength (kg/BMI)	0.66 ± 0.21	0.72 ± 0.20	0.60 ± 0.20	0.18

Data represented as mean and standard deviation. *p*-values < 0.05 are considered statistically significant. BMI: body mass index, SBP: systolic blood pressure, DBP: diastolic blood pressure, TGs: triglycerides, HDL-c: high-density lipids, 6MWt: six-minute walking test.

**Table 2 ijerph-18-09294-t002:** Relationship between sleep quality score and anthropometric, metabolic and fitness parameters.

	Total (*n* = 26)
Variables	rho (*p*-Value)
Age (y)	0.17 (0.39)
Body mass (kg)	0.32 (0.11)
Height (m)	−0.09 (0.65)
Waist circumference (cm)	0.14 (0.47)
Body fat (%)	0.28 (0.15)
Clinical-Metabolic	
SBP (mmHg)	−0.07 (0.69)
DBP (mmHg)	0.03 (0.85)
Fasting glucose (mg/dL)	0.11 (0.58)
HDL-c (mg/dL)	−0.10 (0.62)
Fitness	
Handgrip strength (kg)	−0.01 (0.93)
Relative handgrip strength (kg/BMI)	−0.23 (0.24)

Data represent spearman correlation coefficients (rho). *p*-values < 0.05 are considered statistically significant. SBP: systolic blood pressure, DBP: diastolic blood pressure, HDL-c: high-density lipids.

**Table 3 ijerph-18-09294-t003:** Frequency of parameters related to MetS markers according to sleep quality.

		Sleep Quality		
Parameters	Total (*n* = 26)	Good (GSQ, *n* = 14)	Poor (PSQ, *n* = 12)	*p*-Value
Obesity grade				0.034
Obesity	10 (38.5)	8 (57.14)	2 (16.67)	
Morbid obesity	16 (61.5)	6 (42.86)	10 (83.33)	
Body fat ≥ 48.2%				0.045
No	12 (46.15)	9 (64.29)	3 (25.0)	
Yes	14 (53.85)	5 (35.71)	9 (75.0)	
Clinical-Metabolic				
Fasting glucose ≥ 100 mg/dL				0.054
No	16 (61.54)	11 (78.57)	5 (41.67)	
Yes	10 (38.46)	3 (21.43)	7 (58.33)	
TG ≥ 150 mg/dL				0.25
No	20 (76.92)	12 (85.71)	8 (66.67)	
Yes	6 (23.08)	2 (14.29)	4 (33.33)	
HDL-c < 50 mg/dL				0.43
No	13 (50.0)	6 (42.86)	7 (58.33)	
Yes	13 (50.0)	8 (57.14)	5 (41.67)	
Hypertension DBP/SBP ≥ 90/140 mmHg				0.89
No	9 (34.62)	5 (35.71)	4 (33.33)	
Yes	17 (65.38)	9 (64.29)	8 (66.67)	
Syndrome metabolic parameters				0.36
1	6 (23.08)	3 (21.43)	3 (25.0)	
2	4 (15.38)	3 (21.43)	1 (8.33)	
3	7 (26.92)	5 (35.71)	2 (16.67)	
≥4	9 (34.62)	3 (21.43)	6 (50.0)	

Data represent *n* and proportion (%). *p*-values < 0.05 are considered statistically significant. TGs: triglycerides, HDL-c: high-density lipids, SBP: systolic blood pressure, DBP: diastolic blood pressure.

**Table 4 ijerph-18-09294-t004:** Association between poor sleep quality with MetS markers and physical status.

Comorbidities	OR (95%CI), *p*-Value	OR^Adjusted^ (95%CI), *p*-Value
	Model 0	Model 1
Morbid obesity	6.66 (1.04–42.43), 0.045	8.44 (1.15–66.0), 0.036
Body fat (%)	5.39 (0.98–29.66), 0.05	8.39 (1.13–62.14), 0.037
Glucose ≥ 100 mg/dL	5.13 (0.92–28.57), 0.062	5.71 (0.95–34.05), 0.056
TG ≥ 150 mg/dL	3.0 (0.44–20.43), 0.26	4.79 (0.55–41.61), 0.15
HDL-c < 50 mg/dL	0.53 (0.11–2.55), 0.43	0.63 (0.10–3.98), 0.62
Hypertension	1.11 (0.21–5.63), 0.89	1.14 (0.22–5.91), 0.87
Number of MetS parameters		
1	1.0	1.0
2	0.33 (0.02–5.32), 0.43	0.14 (0.006–2.87), 0.20
3	0.40 (0.04–3.95), 0.43	0.42 (0.03–5.48), 0.51
≥4	2.0 (0.24–16.6), 0.52	5.21 (0.38–70.3), 0.21
MetS	2.42 (0.36–15.94), 0.35	1.5 (0.30–7.43), 0.62
Fitness		
*6MWT ≤ 513.07*	3.6 (0.70–18.25), 0.12	3.40 (0.65–17.56), 0.14
*HGS ≤ 27.86*	1.0 (0.21–4.67), 0.99	0.83 (0.16–4.27), 0.82
*HGS^Relative^ ≤ 0.66*	12.5 (1.85–84.44), 0.010	12.2 (1.79–83.09),0.011

The data shown represent odds ratios (ORs) (Confidence interval, 95% CI) and *p*-values. *p*-values < 0.05 are considered statistically significant. OR adjusted by age. TGs: triglycerides, HDL-c: high-density lipids, 6MWt: six-minute walking test, HGS = handgrip strength.

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
