# Peer review of "Syndrome Metabolic Markers, Fitness and Body Fat Is Associated with Sleep Quality in Women with Severe/Morbid Obesity"

_ijerph, 2021, doi:10.3390/ijerph18179294_

Round 1

Reviewer 1 Report

The study examined the association between sleep quality and metabolic syndrome (MetS) markers, fitness and body fat in women with morbid obesity. The manuscript is well written, however there are specific comments that should be addressed.

Introduction:

-Line 58-60: “…  women present higher levels of inflammation;” There is no references for this sentence, please include it.

-What is the reason for conducting the study only in women? It was not properly explained in the introduction. If there are some stats that show higher prevalence of morbid obesity and/or higher inflammation in women, include that.

Method:

-Line 68-69: Although you mentioned a reference (22) for standard criteria for morbid obesity, it would be valuable to include the obesity classification of your participants. It seems that the participants in GSQ and PSQ are mixed of obesity class I (BMI = 30-35) to super obesity (BMI over 45 and 50). Please clarify that in the method.

Result:

-Line 138: BSQ should be changed to PSQ.

-Knowing that the menopausal transition has impacts on sleep quality and duration, it is important to provide information on menopausal status of the participants specially that you have a wide age range (18-60 y). It can be considered a covariate or a fix parameter and must also be included in regression models.  

-Table 1: How was Body fat (%) measured and/or calculated? Please include it in the method.

-You need to clarify the comparison of parameters is based on the SQ (GSQ vs PQS) in the method section (data analysis).

-Is there any information on diet, physical activity level, marital and socioeconomic status of the participants (education, income, ….)? Are the models adjusted for these covariates?

-Line 94-95: “… the metabolic outcomes total cholesterol (Tc), and low-density lipids (LDL-c) were .. “. There is no information on Tc and LDL-c in the tables.

-Table 4: ≥4 MetS parameters      Does it mean you considered Tc and LDL-c as well? If it is the case that needs to be explained in the footnote.  

Author Response

Reviewer 1

Comment: The study examined the association between sleep quality and metabolic syndrome (MetS) markers, fitness and body fat in women with morbid obesity. The manuscript is well written, however there are specific comments that should be addressed.

Response: Dear reviewer, thanks very much for your help and suggestions, we are sure that the article will improve a lot.

Introduction:

Comment

-Line 58-60: “…  women present higher levels of inflammation;” There is no references for this sentence, please include it.

Response: Sorry about this sentence, we have deleted “…  women present higher levels of inflammation.

We have changed for: Accordingly, patients with morbid obesity present more adverse MetS factors and higher levels of inflammation [5,6,8]

Comment

-What is the reason for conducting the study only in women? It was not properly explained in the introduction. If there are some stats that show higher prevalence of morbid obesity and/or higher inflammation in women, include that.

Response: We included  “besides  patients, especially women suffering from more severe obstructive sleep apnea syndrome and worse sleep quality[8].; however, how MetS risk factors are related to sleep quality in women with morbid obesity needs to be studied deeply. Therefore, the present study aimed to determine the association between sleep quality and MetS markers, such as WC, hypertension, elevated TG, low HDL-C, hyperglycemia, fitness and body fat in women with morbid obesity.

Comment

Method:

-Line 68-69: Although you mentioned a reference (22) for standard criteria for morbid obesity, it would be valuable to include the obesity classification of your participants. It seems that the participants in GSQ and PSQ are mixed of obesity class I (BMI = 30-35) to super obesity (BMI over 45 and 50). Please clarify that in the method.

Response: Done, we have changed the references and clarified it.

Result:

-Line 138: BSQ should be changed to PSQ.

Response: done

Comment

-Knowing that the menopausal transition has impacts on sleep quality and duration, it is important to provide information on menopausal status of the participants specially that you have a wide age range (18-60 y). It can be considered a covariate or a fix parameter and must also be included in regression models.

Response: Dear reviewer, unfortunately, we have not considered this variable… We have added this information to the limitations section.   However, the model was adjusted by age.

Limitations

The main limitation of this study was that the participants’ lifestyle such as physical activity levels and eating habits were not measured. Moreover, the menopausal status and transition were not considered and it could have an impact on sleep quality and duration.

Comment

We also have added on the footnote of the table:  The OR is adjusted by age

-Table 1: How was Body fat (%) measured and/or calculated? Please include it in the method.

Response: we have added the sentence as follow:

2.2.3. Body Composition and Anthropometric Parameters

Body mass (kg) and body fat (%) were measured using a digital bioimpedance scale (TANITATM, model 331, Tokyo, Japan). Height (m) was measured with a SECATM stadiometer (model 214, Hamburg, Germany), with subjects wearing light clothing and without shoes. BMI was calculated as the bodyweight divided by height squared (kg/m2) and then used to estimate the degree of obesity using standard criteria for obesity and morbid obesity classification [22].

Comment

-You need to clarify the comparison of parameters is based on the SQ (GSQ vs PQS) in the method section (data analysis).

Response: we have changed the explanation by:

Their sum builds the global PSQI report, which provides an ‘inverse score’, where a score < 5 denotes good SQ (GSQ) and a score >5 denotes poor SQ (PSQ).

Comment

-Is there any information on diet, physical activity level, marital and socioeconomic status of the participants (education, income, ….)? Are the models adjusted for these covariates?

Response: we apologize, we analyzed the relationship between sleep quality and fitness only

Comment

-Line 94-95: “… the metabolic outcomes total cholesterol (Tc), and low-density lipids (LDL-c) were .. “. There is no information on Tc and LDL-c in the tables.

Response: Sorry about that, we committed a mistake… we have deleted: As additional markers, the metabolic outcomes total cholesterol (Tc) and low-density lipids (LDL-c) were included

Comment

-Table 4: ≥4 MetS parameters.     Does it mean you considered Tc and LDL-c as well? If it is the case, that needs to be explained in the footnote. 

Response: We have deleted both Tc and LDL-c in material and methods section because they are not considered as MetS markers.

Reviewer 2 Report

It is a nice study reporting the association of sleep quality with metabolic syndrome in women. Study is well deigned and presented. Results are nicely interpreted and succinctly written. 

One suggestion is to change the correlation table into figures if possible. As a reader, one would want to see the main effect clearly and figures are always more impactful than tables.

Also in discussion, it would be nice to add sex of the participants when referring to the previous literature. Although sex has been mentioned at some places but not others. Its nice to compare or at least provide the information.

Also, one thing author should add in the methods/aims sections is to why only women were targeted in the present study? Are they historically found to be more susceptible to these disorders or it is more to do with the recruitment? Either way, its important to mention this.

Author Response

Comment

It is a nice study reporting the association of sleep quality with metabolic syndrome in women. Study is well deigned and presented. Results are nicely interpreted and succinctly written.

Response: Dear reviewer, We appreciate your comments and suggestions, we are sure that the manuscript  will improve a lot.

Comment

One suggestion is to change the correlation table into figures if possible. As a reader, one would want to see the main effect clearly and figures are always more impactful than tables.

Response: done, we have added Figure 1 at significant relationship variables.

Also in discussion, it would be nice to add sex of the participants when referring to the previous literature. Although sex has been mentioned at some places but not others. Its nice to compare or at least provide the

Response: we included “besides patients, especially women suffering from more severe obstructive sleep apnea syndrome and worse sleep quality [8].; however, how MetS risk factors are related to sleep quality in women with morbid obesity needs to be studied deeply. Therefore, the present study aimed to determine the association between sleep quality and MetS markers, such as WC, hypertension, elevated TG, low HDL-C, hyperglycemia, fitness and body fat in women with morbid obesity.”

Comment

Also, one thing author should add in the methods/aims sections is to why only women were targeted in the present study? Are they historically found to be more susceptible to these disorders or it is more to do with the recruitment? Either way, its important to mention this.

Response: Dear reviewer in the rationale of the study we have added:

Introduction:

besides of, patients especially women suffering from more severe obstructive sleep apnea syndrome and worse sleep quality [8].;

Discussion

Another study conducted in women who were overweight or obese reported that the participants who had worse sleep quality obtained significantly higher HOMA2-IR; therefore, the authors concluded that more studies were needed to establish whether enhancing sleep quality improves insulin resistance [35], besides of, the evidence has shown that women have lower sleep quality than men [8].

Round 2

Reviewer 1 Report

no further comments